# Phytochemical Analysis of *Maerua oblongifolia,* and Assessment of the Genetic Stability of *M. oblongifolia* under In Vitro Nanoparticles Exposure

**Hassan O. Shaikhaldein** *, **Fahad Al-Qurainy, Mohamed Tarroum** , **Salim Khan, Mohammad Nadeem** and **Abdalrhaman M. Salih**

Botany and Microbiology Department, College of Science, King Saud University,
P.O. Box 2455, Riyadh 11451, Saudi Arabia; falqurainy@ksu.edu.sa (F.A.-Q.); mtarroum@ksu.edu.sa (M.T.);
skhan2@ksu.edu.sa (S.K.); mnadeem@ksu.edu.sa (M.N.); abdalrahamanm@gmail.com (A.M.S.)
* Correspondence: hassanbb2@gmail.com or hshaikhaldein@ksu.edu.sa; Tel.: +966-531512830

**Abstract:** *Maerua oblongifolia* (Forssk.) is a rare medicinal plant in Saudi Arabia that is threatened with extinction owing to overexploitation, climate change, and poor seed germination. This study aimed to identify, for the first time, the phytochemical compounds existing in *M. oblongifolia* leaves' extract using gas chromatography and mass spectroscopy (GC-MS). In addition, it aimed to determine the plant growth and genetic uniformity of the plant under the exposure of in vitro biogenic silver and zinc oxide nanoparticles. The GC-MS analysis detected 28 phytochemical compounds. The main compounds obtained from the leaf extracts were triphenylphosphine oxide and 4,5-Dihydrooxazole-5-one, 2-methyl-4-[2,3,4-methozxybenzylidnen]-. The supplementation of AgNPs and ZnO NPs to the culture media significantly enhanced the plant biomass, shoot length, and shoot regeneration of *M. oblongifolia*. The genetic stability of the plant material was evaluated using inter-simple sequence repeat (ISSR) markers. The application of Ag and ZnO NPs showed genetic stability among treated plants. However, the higher concentration of both nanoparticles induced minor genetic variations recorded as 4.4 and 2.2% in Ag and ZnO NPs, respectively. This work focused on the detection of phytochemical active constituents from *M. oblongifolia* shoot cultures, and it will be useful for the large-scale manufacturing of these compounds for pharmaceutical and commercial purposes. In addition, it confirmed that the exposure of silver and zinc oxide nanoparticles to the in vitro culture media of plant tissues might be a secure technique with which to produce true-to-type plants.

**Keywords:** *Maerua oblongifolia*; phytochemical compounds; nanoparticles; genetic stability

## 1. Introduction

*Maerua oblongifolia* is a rare medicinal plant found in Saudi Arabia that belongs to the family Capparaceae. Like other medicinal plants, this plant is widely used in traditional herbal healing practices for the medication of both humans and animals [1]. Medicinally, the plant is utilized as an antimicrobial agent as well as for curing several health disorders such as stomach aches, typhoid, fever, skin infections, urinary calculi, and diabetes mellitus [2]. Traditionally and commercially, *M. oblongifolia* is used to prepare numerous herbal formulations, however, research is still needed, as there is no scientific evidence to prove its potential medicinal properties [3]. Therefore, knowledge of the chemical ingredients of traditional medicinal plants like *M. oblongifolia* is very important because such information will be of value for the synthesis of complex chemical substances, as well as the identification of new sources of natural medication products.

Local informants and herbalists both indicated that the wild populations of this species has declined, owing to the overexploitation of the plant as a medicine, timber, food, and fodder, as well as its depressed regeneration rate. Therefore, enhancing the *M. oblongifolia*

plant regeneration through effective techniques like tissue culture is necessary [4,5]. This can be achieved effectively with the application of nanoparticles (NPs).

The increasing attention given to NPs results in their wide application in numerous disciplines, including plant science [6]. Various types of NPs have been developed for plants, such as zinc, silver, gold, titanium, copper, magnesium, and silicon NPs. However, silver and zinc oxide nanoparticles are recognized as being the most researched for plant research applications. It was reported that AgNPs and ZnO NPs can affect the regeneration capacity of several in vitro raised plant species [7].

Genotoxicity can be caused directly or indirectly by nanoparticles. In the direct mechanism, nanoparticles interact either mechanically or chemically with the DNA by passing through the membrane of the cell and nucleus. On the other hand, the indirect mechanism occurs due to interactions with nuclear proteins (engaged in replication, transcription, and translation) or induced oxidative stress. [8]. In *Vicia faba* L. root-tips treated with AgNPs, chromosomal aberrations such as chromatin bridges, stickiness, disrupted meta-phase, and chromosomal breaks impacting mitosis were observed. [9]. Thus, nanoparticles may have an impact on plant development, metabolism, and phenotypic plasticity.

In general, there is a scarcity of thorough research proving the (cyto) genetic, biochemical, and phenotypic effects of NPs on plants, particularly when applied to in vitro grown plants. In addition, keeping the original genetic background of medicinal plants during in vitro propagation is very important to protect valuable endemic genotypes/ecotypes, particularly if exposed to different types of NPs. The aims of this study were to identify the phytochemical compounds in the extracts of the *M. oblongifolia* leaves and to verify the genetic stability of *M. oblongifolia* treated in vitro with AgNPs and ZnO NPs.

## 2. Materials and Methods

### 2.1. Plant Materials

Specimens of wild *M. oblongifolia* were brought from Jazan, Saudi Arabia and propagated in vitro through micropropagation in Murashige and Skoog (MS) media, as per the protocol reported by Al-Qurainy et al. [5] in King Saud University tissue culture lab.

### 2.2. Gas Chromatography-Mass Spectrometry Analysis

Analysis using gas chromatography and mass spectrometry was conducted using (QP2010 Ultra, Shimadzu, Tokyo, Japan). An Rtx-5MS column (30 m; 0.25 mm i.d.; 0.25 m) was used. The carrier gas applied was helium at a flow rate of 1.6 mL/min. The oven temperature was programmed initially at 50 °C for 3 min, then programmed to climb to 280 °C at a rate of 10 °C/min, hold for 3 min, then increase to 300 °C at a rate of 2 °C/min, and hold for 10 min. The injector and detector temperatures were set to 250 and 275 degrees Celsius, respectively. MS was used to identify the organic phase, which was conducted in scan mode at 70 eV and electron ionization mode. The mass spectra obtained were compared with the database saved in the NIST reference spectra library.

### 2.3. AgNPs Treatment

*Ochradenus arabicus* Chaudhary, Hillc. & A.G.Mill. leaves were used for the synthesis of AgNPs biologically. The detailed characteristics and synthesis procedure of nanoparticles are given in Shaikhaldein et al. [7].

To prepare the AgNPs, 100-mL solution of 1 mM $AgNO_3$ was added to the plant *Ochradenus arabicus* leaf extract in a ratio of 1:1. Reduction of $Ag^+$ in the reaction mixture was observed by the change in color of the reaction solution from light yellow to brown, indicating the initial formation of AgNPs (***S 1***). The particle morphology and size of the phytomediated AgNPs were photographed using transmission electron microscopy (TEM).

Biologically synthesized AgNPs in different concentrations (0 mg, 10 mg, 20 mg, 30 mg, 40 mg, and 50 mg $L^{-1}$) were added to afford 50 mL of MS media in each Magenta box (GA-7). *M. oblongifolia* nodal segments of 2–3 cm in length were transplanted in the boxes. Every box contained five explants.

### 2.4. ZnO NPs Treatment

Green synthesized ZnO NPs using *Ochradenus arabicus* leaves were prepared as reported by Shaikhaldein et al. [10]. The constituents of the preparation of ZnO NPs included zinc acetate, sodium hydroxide, and *Ochradenus arabicus* leaf extract. To prepare ZnO NPs, 100 mL of leaf extract was added to 2 mM zinc acetate and 0.2 mM of sodium hydroxide, followed by continuous stirring during day and night. The mixture's color changed to a pale yellow, the first evidence of ZnO NPs formation. The precipitate was then removed from the reaction solution and dried in a hot air oven using Whatman number-1 filter paper. The pellet was washed with milli q water. Then, the pellet was washed with absolute alcohol and dried at 60 °C for 24 h. To obtain the final product of the biosynthesized ZnO NPs, the powder was calcined for 3 h at 600 °C in a furnace. A simplified scheme for the preparation of ZnO NPs is shown in (*S 2*). The biosynthesized ZnO NPs were confirmed using TEM to determine their shape and size.

The green synthesized ZnO NPs were augmented to the MS Media and concentrations of (0, 1.25, 2.5, 5, 10, or 20 mg L$^{-1}$) were used. The treatment was performed with the same description explained for AgNPs treatment.

After 6 weeks of culture, samples were collected to evaluate the effect of the application of Ag and Zn O nanoparticles on in vitro propagated *M. oblongifolia*, as well as assess the genetic stability of nanoparticle-treated plants using inter-simple sequence repeats (ISSR) markers. ISSR is a relatively new technology that has proven to be a powerful, quick, easy, re-producible, and low-cost tool to assess genetic diversity in a variety of plant species.

### 2.5. Transmission Electron Microscopy (TEM)

To determine the size and morphology of Ag and ZnO via TEM, the samples were suspended in chloroform and put onto a carbon-coated nickel micro grid, which was then air-dried in a fume hood. The TEM visualizations were carried out on a JEOL 2100EX microscope with a 100 kV accelerating voltage.

### 2.6. DNA Isolation and Genetic Stability Assessment Using ISSR Markers

For genetic stability assessment, approximately 200 mg of fresh shoots (shoots control, or shoot treated with different concentrations of either Ag or ZnO NPs) as well as mother plant (MP) were submerged in liquid nitrogen for DNA isolation using the CTAB (Cetyl Trimethyl Ammonium Bromide) method as reported by Doyle and Doyle [11]. The extracted DNA was analyzed for its quality standards via a Nanodrop spectrophotometer (Nanodrop 2000, Thermo Scientific, Waltham, MA, USA). The quality and purity of the DNA were also confirmed by running on 1% agarose gel electrophoresis. All of the genomic DNA samples were then diluted to a final concentration of 50 ng/μL with 1X TE buffer (10 mM Tris–HCl; pH 8.0; 1 mM EDTA). DNA samples were placed in a deep-freezer at −20 °C for further use.

Based on the clear and accurate amplified band profiles, a set of 7 primers was selected to assess the genetic uniformity among the mother plant, and control and NPs-treated shoots. A PCR thermal cycler (Cyber Lab, NJ, USA) was used to program the ISSR amplification reaction. The PCR reaction mixture, consisting of 10 PCR mix with a tracking dye bromophenol green, 2 μL primer, 50 ng extracted DNA, and the appropriate volume of milli q water, had a total volume of 20 μL. The following amplification profile was used: initial denaturation at 94 °C for 5 min, followed by 35 cycles consisting of denaturation at 94 °C for 1 min, annealing temperature varying between 47 to 50 °C for 1 min, extension at 72 °C for 1.5 min, and a final extension step at 72 °C for 7 min. Finally, after the PCR cycles were completed, the DNA amplicons were stored at −20 °C until further investigation.

### 2.7. Electrophoretic Analysis of PCR-Amplified Products

The reaction products were electrophoresed on 1.2 percent agarose gels, stained with ethidium bromide, and shot with a digital camera with UV filter adaptor under UV transilluminator using Bio-Rad documentation gel system (Bio-Rad Laboratories Inc., Hercules,

CA, USA). As a molecular marker for band size, a 1000 bp DNA ladder (Pharmacia) was used. The presence (1) or absence (0) of bands at specific places on the gel characterized each amplified band profile. Genetic uniformity coefficients among shoots of the mother plant, and control and AgNPs-treated shoots were estimated using Jaccard's coefficient [12]. The cluster analysis was performed with NTSYSpc 2.1 software via the unweighted pair–group method with arithmetic mean (UPGMA) algorithm and the similarity matrix [13].

### 2.8. Statistical Analyses

The experiments were performed in a completely randomized design. Data were verified statistically by using one-way analysis of variance, and means for each treatment were evaluated with the Duncan's new multiple range test ($p \leq 0.05$) in SPSS v. 20 for Windows.

### 3. Results

### 3.1. GC-MS Analysis

The gas chromatography and mass spectroscopy (GC-MS) studies of methanolic extracts of *M. oblongifolia* leaves revealed 28 phytochemical compounds (Table 1 and Figure 1). Based on the peak area percentage, the major compounds found in the extract were triphenylphosphine oxide (16.36) and 4,5-Dihydrooxazole-5-one, 2-methyl-4-[2,3,4-methozxybenzylidnen] (11.75). One unknown compound (p-Methoxybenzylazidoformate) was also identified.

**Table 1.** Activity of phytocomponents identified in the methanolic extract of *M. oblongifolia* leaf.

| | Compounds | RT | % | Bioactivity | References |
|---|---|---|---|---|---|
| 1 | p-Methoxybenzylazidoformate | 4.381 | 1.13 | Unknown | |
| 2 | Undecane | 9.59 | 1.88 | Alarm pheromone of the ant *Componotus obscuripes* | [14] |
| 3 | Eicosane | 9.669 | 2.02 | Antifungal and Antibacterial activities | [15] |
| 4 | Cyclotrisiloxane hexamethyl | 11.02 | 2.32 | Antimicrobial activities | [16] |
| 5 | Phthalic Acid Esters | 30.88 | 2.69 | Antimicrobial, allelopathic, Insecticidal activities | [17] |
| 6 | Cyclononasiloxane octadecamethyl | 38.50 | 1.36 | Antifouling, Antimicrobial and antioxidant activities | [15] |
| 7 | Piperidine, 1-(5-trifluoromethyl-2-pyridyl)-4-(1H-pyrrol-1-yl)- | 39.6 | 1.52 | Antibacterial activities | [18] |
| 8 | Phenol, 2,2'-methylenebis [6-(1,1-dimethylethyl)-4-methyl- | 40.41 | 6.99 | Antibacterial activity | [19] |
| 9 | Octasiloxane, 1,1,3,3,5,5,7,7,9,9,11,11,13,13,15,15-hexadecamethyl- | 40.54 | 1.70 | Anti-bacteria1activity | [20] |
| 10 | 1,2-Bis(diphenylphosphino)benzene | 40.89 | 4.94 | Antileukemia activity | [21] |
| 11 | 4,5-Dihydrooxazole-5-one, 2-methyl-4-[2,3,4-methozxybenzylidnen]- | 41.1 | 11.75 | Antimalarial agents | [22] |
| 12 | Propiophenone, 2'-(trimethylsiloxy)- | 41.24 | 3.99 | Antibacterial activity | [23] |
| 13 | Triphenylphosphine oxide | 41.55 | 16.36 | Crystallization aid | [24] |
| 14 | Heptasiloxane, 1,1,3,3,5,5,7,7,9,9,11,11,13,13-tetradecamethyl- | 42.07 | 4.16 | Anti-diabetic activities | [25] |
| 15 | 1H-Indole-2-carboxylic acid, 6-(4-ethoxyphenyl)-3-methyl-4-oxo-4,5,6,7-tetrahydro-, isopropyl ester | 42.2 | 1.61 | Anti-diabetic activities | [26] |
| 16 | 1, 2-Benzisothiazol-3-amine tbdms, | 42.40 | 1.17 | Antimicrobial and antioxidant activities | [27] |
| 17 | 1-Methyl-3-phenylindole | 43.37 | 1.58 | Used in colorimetric assay of lipid peroxidation | [20] |
| 18 | Trimethyl [4-(2-methyl-4-oxo-2-pentyl)phenoxy]silane | 43.58 | 5.51 | Antioxidant, antibacteria1, anti-inflammatory | [28] |
| 19 | 5-Methyl-2-trimethylsilyloxy-acetophenone | 43.69 | 1.34 | Antioxidant activity | [29] |

**Table 1.** *Cont.*

| | Compounds | RT | % | Bioactivity | References |
|---|---|---|---|---|---|
| 20 | 1H-Indole, 1-methyl-2-phenyl- | 43.84 | 1.38 | Antibacterial, antifungal, antitubercular and antitumor properties | [30,31] |
| 21 | Tetrasiloxane, decamethyl- | 44.01 | 1.44 | Antifungal activity | [32] |
| 22 | 4-Methyl-2-trimethylsilyloxy-acetophenone | 44.26 | 1.54 | Anti-Listeria activity | [33] |
| 23 | Benzene, 2-[(tert-butyldimethylsilyl)oxy]-1-isopropyl-4-methyl- | 44.37 | 2.43 | Antibacterial activity | [34] |
| 24 | 1,4-Phthalazinedione, 2,3-dihydro-6-nitro- | 44.47 | 1.37 | Derivatives have vasorelaxant activity and antibacterial activity | [35,36] |
| 25 | Methyltris(trimethylsiloxy)silane | 44.51 | 1.95 | Antibacterial activity | [37] |
| 26 | Silane, 1,4-phenylenebis[trimethyl- | 44.83 | 4.40 | Antimicrobial | [38] |
| 27 | 2,4,6-Cycloheptatrien-1-one, 3,5-bis-trimethylsilyl- | 44.99 | 6.03 | Anti-anaemic | [39] |
| 28 | Benzo [h] quinoline, 2,4-dimethyl- | 52.4 | 5.43 | Antimalarial activity | [40] |

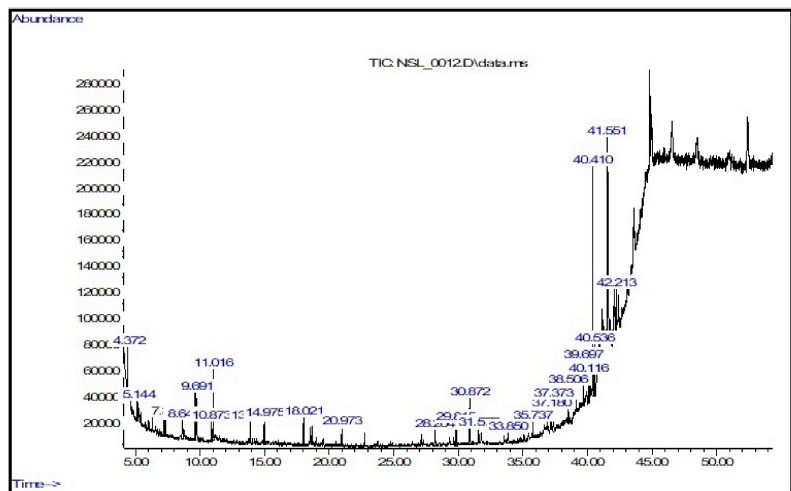

**Figure 1.** GC–MS traces of phytochemical constituents of *M. oblongifolia* methanolic extract.

*3.2. Transmission Electron Microscopy (TEM)*

The TEM image of the AgNPs is shown in Figure 2. The size of biosynthesized AgNPs ranged between 11–30 nm, and they were spherical in shape. On the other hand, the shape of biosynthesized ZnO NPs was hexagonal, according to the TEM image, and the size was found to be below 100 nm (Figure 3).

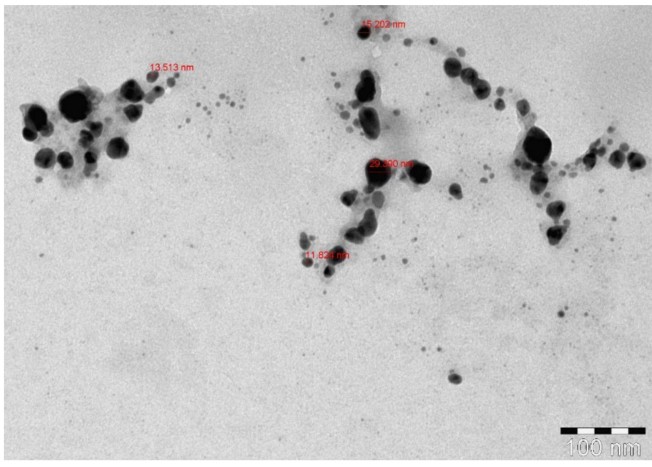

**Figure 2.** TEM micrograph of biosynthesized AgNPs.

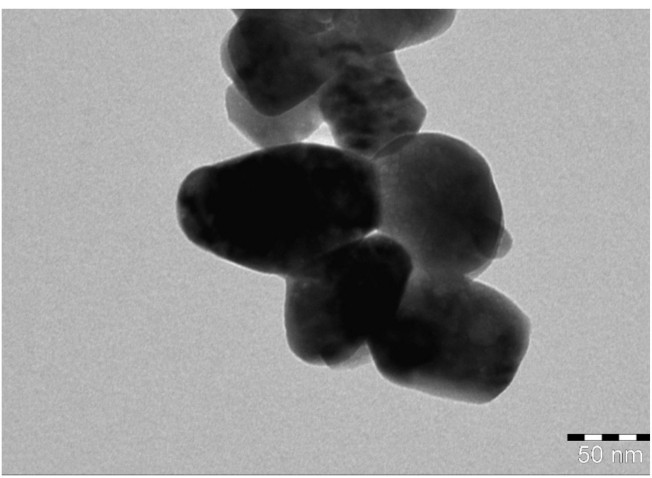

**Figure 3.** TEM micrograph of biosynthesized ZnO NPs.

### 3.3. Impacts of AgNPs on Shoot Growth

The impact of AgNPs on the developmental characteristics (shoot number, shoot length, fresh weight, and dry weight) of the plant are shown in Figure 4. It was observed that with the increase in Ag NPs concentrations, all growth characteristics also increased. However, after a certain concentration (20 mg/L), the numbers of shoot, shoot length, and plant biomass were found to decline.

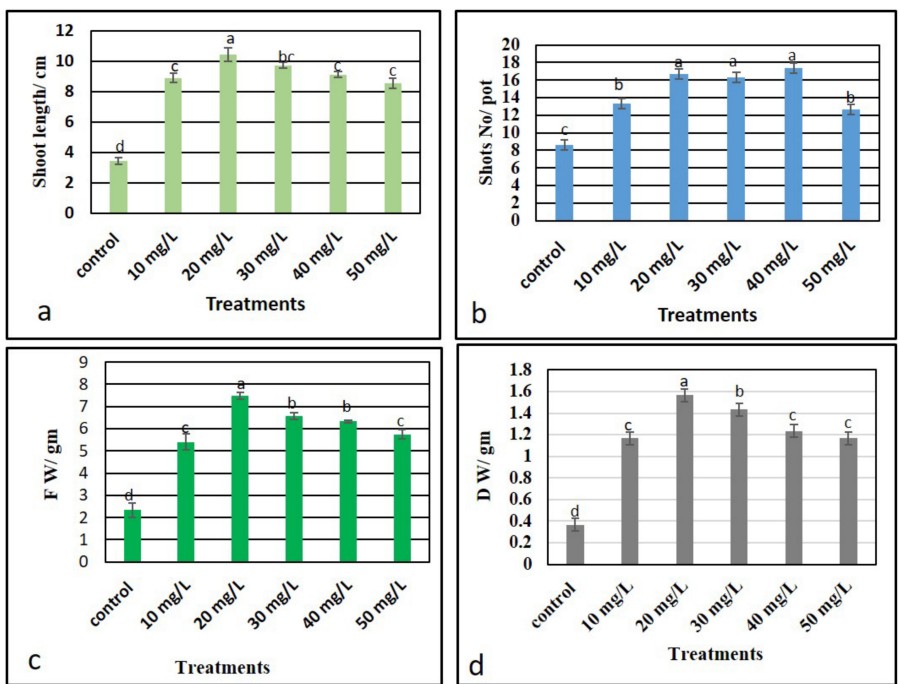

**Figure 4.** Influence of AgNPs on the in vitro regeneration of *M. oblongifolia* after 45 days of treatment in MS media: shoot length (**a**); shoot number (**b**); fresh weight (**c**); and dry weight (**d**). Means $\pm$ SD for each treatment followed by the same letters are not significantly different according to one-way analysis of variance ($p \leq 0.05$).

### 3.4. Impacts of ZnO NPs on Shoot Growth

The supplementation of ZnO NPs to *M. oblongifolia* shoots significantly increased all morphological attributes, including plant biomass (fresh and dry weights) shoot number, shoot length, and leaf number (Figure 5). All of the developmental parameters were boosted

in the plants treated with 5 mg/L ZnO NPs. In contrast, the control group showed the least amount of growth in all morphological characteristics.

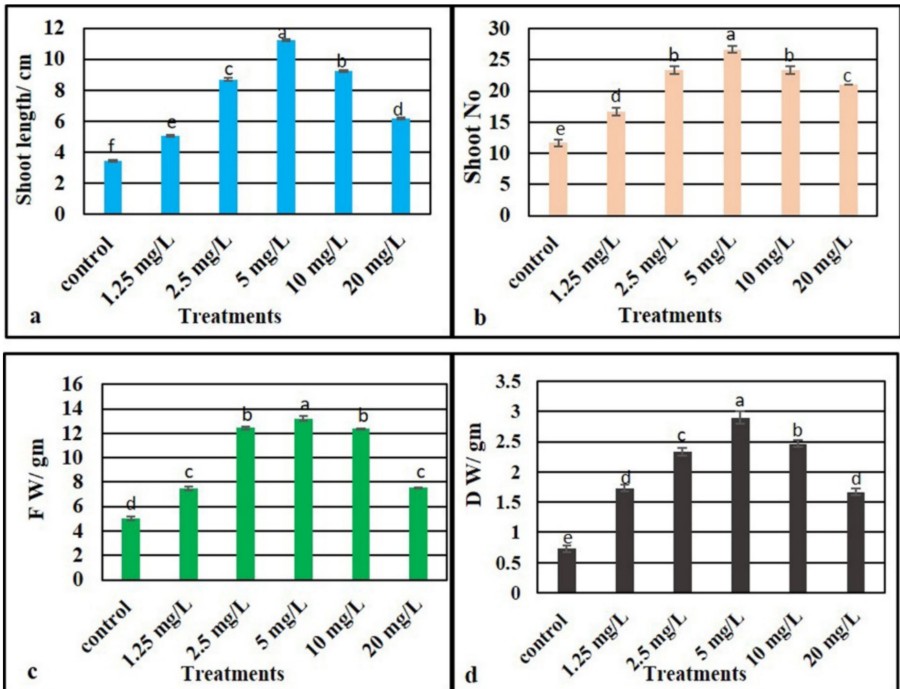

**Figure 5.** Influence of ZnO NPs on the in vitro regeneration of *M. oblongifolia* after 45 days of treatment in MS media: shoot length (**a**); shoot number (**b**); fresh weight (**c**); and dry weight (**d**). Means ± SD for each treatment followed by the same letters are not significantly different according to one-way analysis of variance ($p \leq 0.05$).

*3.5. Detection of Genetic Uniformity Using ISSR (PCR) Molecular Marker*

An inter-simple sequence repeat (ISSR) marker was applied to detect the level of genetic uniformity of *M oblongifolia* shoots exposed to different doses of either AgNPs or ZO NPs. Six ISSR primers (UBC-7, UBC-12, UBC-17, UBC-20, UBC-21 and UBC-24) were picked due to their ability to produce clear and satisfactory amplification bands. The total number of generated amplified fragments using six ISSR primers were 45 in each Ag and ZnO NPs treatment, ranging between 250 to 1300 bps with an average of 6.4 bands per primer; 43 bands in AgNPs were monomorphic, and two bands regenerated polymorphism, which represent only 4.4% DNA polymorphism, as seen in Table 2. On other hand, 44 out of 45 bands in ZnO NPS were monomorphic, whereas only one band regenerated polymorphism, representing 2.2% DNA polymorphism, as seen in Table 3.

**Table 2.** List of primers and their sequences of the amplified fragments generated by ISSR markers for DNA fingerprinting of in vitro *M. oblongifolia* plants under AgNPs exposure.

| Primer Name | Primer Sequence | Total Number of Amplified DNA Bands | Number of Monomorphic Bands per Primer | Number of Polymorphic Bands per Primer | Percent of Monomorphism |
|---|---|---|---|---|---|
| UBC-7 | 5′-AGA GAG AGA GAG AGA GT-3′ | 8 | 8 | 0 | 100 |
| UBC-12 | 5′-GAG AGA GAG AGA GAG AA-3′ | 7 | 6 | 1 | 100 |
| UBC-17 | 5′-CTC TCT CTC TCT CTC TG-3′ | 8 | 7 | 1 | 100 |
| UBC-20 | 5′-CAC ACA CAC ACA CAC AA-3′ | 6 | 6 | 0 | 100 |
| UBC-21 | 5′-GTG TGT GTG TGT GTG TC-3′ | 9 | 9 | 0 | 100 |
| UBC-24 | 5′-GTG TGT GTG TGT GTG TT-3′ | 7 | 7 | 0 | 100 |
| Total | 5′-TCT CTC TCT CTC TCT CG-3′ | 45 | 43 | 2 | 95.5 |

**Table 3.** List of primers and their sequences of the amplified fragments generated by ISSR markers for DNA fingerprinting of in vitro *M. oblongifolia* plants under ZnO NPs exposure.

| Prim Name | Primer Sequence | Total Number of Amplified DNA Bands | Number of Monomorphic Bands per Primer | Number of Polymorphic Bands per Primer | Percent of Monomorphism |
|---|---|---|---|---|---|
| UBC-7 | 5′-AGA GAG AGA GAG AGA GT-3′ | 8 | 8 | 0 | 100 |
| UBC-12 | 5′-GAG AGA GAG AGA GAG AA-3′ | 7 | 6 | 1 | 100 |
| UBC-17 | 5′-CTC TCT CTC TCT CTC TG-3′ | 8 | 8 | 0 | 100 |
| UBC-20 | 5′-CAC ACA CAC ACA CAC AA-3′ | 6 | 6 | 0 | 100 |
| UBC-21 | 5′-GTG TGT GTG TGT GTG TC-3′ | 9 | 9 | 0 | 100 |
| UBC-24 | 5′-GTG TGT GTG TGT GTG TT-3′ | 7 | 7 | 0 | 100 |
| Total | 5′-TCT CTC TCT CTC TCT CG-3′ | 45 | 44 | 1 | 97.8 |

The total number of amplified bands varied from 6 to 9 per primer. UBC-21 primer detected the highest number of amplified (*loci*) products (9), and UBC-20 detected the minimum (*loci*) (6). In the AgNPs-treated plant, only the UBC 12 and UBC 17 primers under the highest AgNPs concentrations (40 and 50 mg/L) detected polymorphic loci. On the other hand, no polymorphisms were found in the other four primers. In the case of *M. oblongifolia* shoots treated with ZnO NPs, the maximum polymorphism was observed in UBC-12 after adding 20 mg/ L of ZnO NPs, whereas no polymorphisms were found in the other five primers, and all the other treatments, including the mother plant, were monomorphic and similar to each other. A monomorphic banding pattern was observed for all the amplified bands among the nanoparticle-treated shoots and mother plant, indicating the absence of variability. Examples of ISSR patterns amplified with primer UBC-17 are presented in Figure 6 for AgNPs and Figure 7 for ZnO NPs.

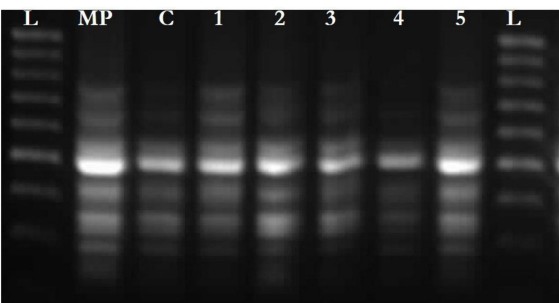

**Figure 6.** A band pattern produced by UBC 17 primer for shoots: Ladder (L), Mother plant (MP), control (C), and *M. oblongifolia* shoots' exposure to AgNPs treatments (1, 2, 3, 4, and 5) represent AgNPs treatment 10, 20, 30, 40, and 50 mg/L, respectively.

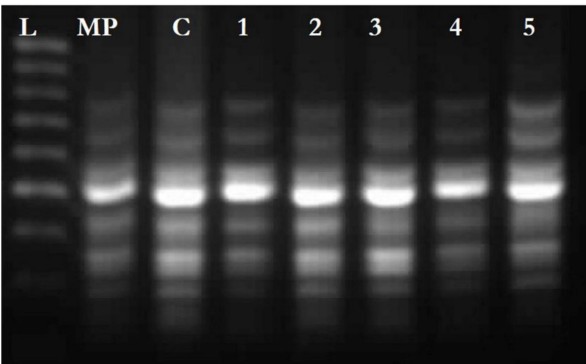

**Figure 7.** A band pattern produced by UBC 17 primer: Ladder (L), Mother plant (MP), control (C) and *M. oblongifolia* shoots exposed to ZnO NPs treatments (1, 2, 3, 4 and 5) represent ZnO NPs treatment 1.25, 2.5, 5, 10, and 20 mg/L, respectively.

### 3.6. Data Analysis

The similarity degree based on Jaccard's similarity coefficient between the nanoparticle-treated plants and mother plant was evaluated. AgNPs with different treatments (0, 10, 20, 30, 40 and 50 mg/L) showed similarity ranging from 0.95 to 1.00, as presented in (**S 3**). The highest similarity value (1.00) was observed between all AgNPs-treated shoots with the mother plant, whereas the shoots treated with the highest concentrations of AgNPs (40 and 50 mg/L) demonstrated the lowest value of similarity (0.97 and 0.95, respectively) to the other treatments and the mother plant. As seen in Figure 8, the dendrograms according to a thorough unweighted pair–group method with arithmetic mean (UPGMA) revealed a 97% similarity among regenerated plants with the mother plant. An UPGMA analysis of the ISSR marker system indicated that treatments of 0, 10, 20, and 30 mg/L AgNPs are identical to the MP. Plants treated with 40 and 50 mg/L also clustered close to the MP.

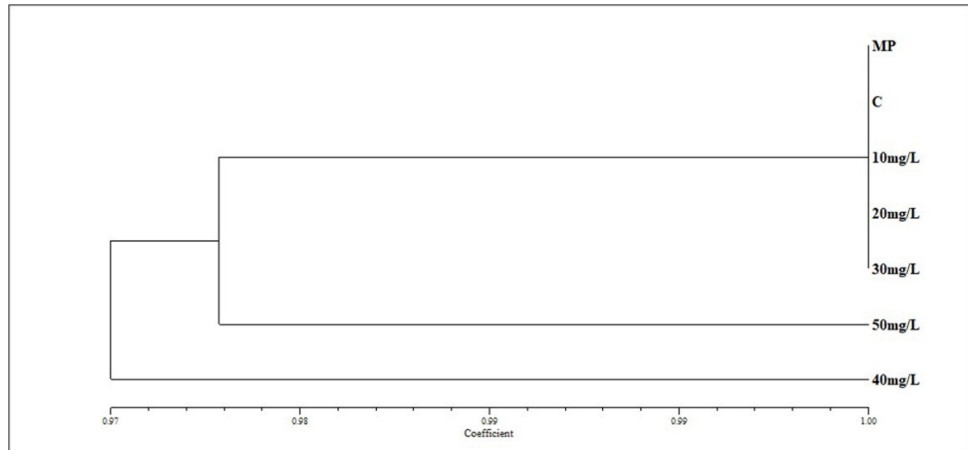

**Figure 8.** Dendrogram based on UPGMA analysis presenting the relationships between *M. oblongifolia* shoots plants treated with AgNPs (C, 10, 20, 30, 40 and 50 mg/L); MP = Mother plant.

In the case of ZnO NPs treatments, the values of uniformity ranged between 0.98 and 1.00 (**S 4**). The highest similarity value (1.00) was observed in all ZnO NPs-treated plants compared to the MP, whereas the shoots treated with 20 mg/L demonstrated the lowest value of similarity (0.98). The dendrogram generated for the ISSR marker showed that with the exception of 5 mg/L of ZnO NPs, the rest of the ZnO NPs-treated plants were 100% identical to the MP. The dendrograms generated through the UPGMA analysis revealed a 98% similarity amongst the ZnO NPs treatment regenerates and the mother plant (Figure 9).

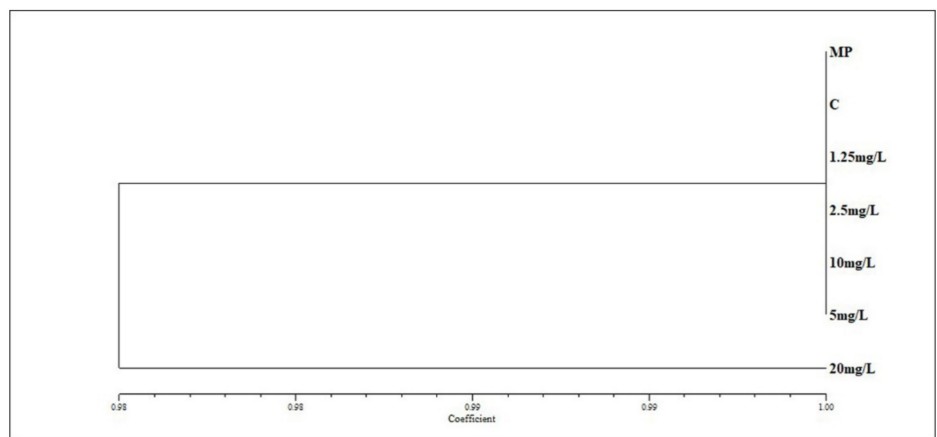

**Figure 9.** Dendrogram based on UPGMA analysis presenting the relationships between *M. oblongifolia* shoots plants treated with ZnO NPs (C, 1.25, 1.5, 5, 10 and 20 mg/L); MP = Mother plant.

## 4. Discussion

The use of medicinal plants in the curing of various human diseases usually depends on their phytochemical compounds. In the present study, the GC/MS analysis showed that 28 phytochemical constituents were present in the methanolic extract of *M. oblongifolia*. The fragmentation pattern of the major compound is triphenylphosphine oxide, the retention time is 41.55, and the peak area percentage is 16.36. The next highest found compound is 4,5-Dihydrooxazole-5-one, 2-methyl-4-[2,3,4-methozxybenzylidnen]- with a retention time of 41.1 and peak area percentage of 11.75. These compounds contribute to activates such as antibacterial, antifungal, antioxidant, anti-listeria, antianemic, antidiabetic, antitumor and many other activates. The presence of antimicrobial, antifungal, antimicrobial, antidiabetic, and chemotaxonomic activates in *M. oblongifolia* plants were previously reported earlier [41–45]. The preliminary phytochemical analysis of a *M. oblongifolia* root extract revealed phytochemicals such as alkaloids, phytosterols, and saponins [46].

Growth parameters such as plant biomass (fresh and dry weights) and shoot number are very important for understanding the impacts of nanoparticles applied to plants. In this study, the growth-improving effects of Ag and ZnO NPs used for the culture medium of *M. oblongifolia* shoots were evaluated. The application of AgNPs significantly enhanced the plant's vegetative growth, including an improvement in plant biomass, plant height, and number of plants. Similar results were obtained for in vitro raised *Stevia rebaudiana* (Bertoni) Bertoni [47] and *Chrysanthemum morifolium* (Ramat.) Hemsl. [48]. The induced growth enhancements caused by different AgNP doses, especially at 20 mg/L, might be due to the role of AgNPs in blocking ethylene signaling in plants [49]. In the case of ZnO NPs, similar result presented in AgNPs were also obtained, where ZnO NPs significantly affected plant biomass, shoot production, and shoot length. The application of 5 mg/L recorded the greatest fresh weight, dry weight, shoot multiplication and shoot length per explant. This increase in plant growth is probably due to the role of Zn in the generation of tryptophan—the precursor of indole-3-acetic acid phytohormone [50]. Furthermore, the phytohormone biosynthesis of cytokinins and gibberellins are found to be modulated by ZnO NPs, these phytohormone can lead to an expansion in the number of internodes per plant [51]. The effectiveness of ZnO NPs in increasing plant growth was also reported in *Brassica juncea* (L.) Czern. [52], tomato [53], and maize [54]. Generally, metallic NPs have already been applied to numerous plants to provide positive impacts on shoot regeneration and in vitro growth [55].

The potentiality of genetic variations occurring during in vitro processes cannot be ruled out because micropropagation methods are known to cause somaclonal variations in plants raised in vitro due to stress induced by nanoparticles' effect. The genetic uniformity of micro-propagated plants, especially those exposed in a basal medium supplemented with either nano-silver or nano-zinc oxide treatment, is to be, therefore, ensured. Thus, a distinguished technique (DNA-based molecular markers) has been developed for the purpose, and therefore, is being applied to different plant species. Among all types of markers, ISSR markers have already proven to be cost effective and dependable, and most reproducible markers depend on inter-simple sequence of repeatable sequences in the genomic DNA of all organisms. Furthermore, the accuracy of the results with ISSR primers is more reliable than other primers, like RFLP and RAPD [56]. In this study, ISSR markers were employed to check for any genetic alteration caused by either AgNPs or ZnO NPs. An ISSR analysis of the *M. oblongifolia* shoots treated with NPs showed almost an identical profile with the control plant as well as with the mother plant, indicating that no genetic variation had occurred after nanoparticles exposure. All ISSR profiles from the treatment with NPs plants were found to be monomorphic and analogous to those of the control and the donor plant. Corresponding to this study, the absence of genetic variation has been stated in AgNPs-treated micropropagated shoots of *Phyllanthus amarus* Schumach. & Thonn [57]. Furthermore, the incidence of monomorphism among the ISSR markers confirmed the genetic homogeneity and stability of cultures raised through Ag and ZnO nanoparticles [58]. Slight polymorphisms were generated in plant shoots at the application

of the highest doses in both Ag (40 and 50 mg/L) and ZnO (20 mg/L) nanoparticles. The obtained low polymorphisms in this study are comparable to that observed in peppermint (*Mentha × piperita* L.) [59] and *Salvadora persica* L. [60]. The polymorphisms that occurred thus might be due to the high concentrations of Ag and ZnO NPs, which caused the genotoxic or mutagenic effects [61]. Genotoxic effects may have occurred due to following mechanisms: (1) high levels of reactive oxygen species induced by high doses of NPs [62]; and (2) levels of gene expression mutated in response to the cellular stress induced by the NPs [63]. The high concentrations of Ag and ZnO NPs may intervene with the delicate cellular homeostatic balance and thus alter complex intracellular signaling pathways, causing a cascade of genotoxic effects [64]. Studies have indicated that different genotoxic effects occur depending on the plant species as well as the size, shape, and concentrations of nanoparticles. The results of this study revealed that the monomorphic bands of ISSR primers showed high uniformity between the *M. oblongifolia* plants under the effect of Ag and ZnO nanoparticles with the mother plant.

## 5. Conclusions

In conclusion, GC-MS screening of *M. Oblongifolia* leaves showed the existence of 28 phytochemical compounds; this indicates that the leaves of *M. Oblongifolia* could be a source of high-value secondary metabolites and justifies the use of the plant for the traditional treatment of various ailments.

The testing of the genetic fidelity of in vitro-raised plants, especially those multiplied in a basal medium supplemented with nanoparticles is inevitable and remains one of the most important prerequisites. Our results indicate that the addition of Ag and ZnO NPs into the culture medium at specific doses stimulates the proliferation of shoots and multiplication ratio in *M. oblongifolia.* An inter-simple sequence repeat (ISSR) analysis on Ag and ZnO NPs exposed to *M. oblongifolia* shoots showed genetic uniformity among treated plants. However, the higher doses of the nanoparticle treatments induced low polymorphism, confirming minor genotoxic effects on the plant.

Based on the results obtained in the present study, it can be concluded that the application of silver and zinc oxide nanoparticles to the in vitro culture media of plant tissues might be a safe technique to produce true-to-type plants.

**Author Contributions:** Conceptualization, H.O.S.; Data curation, H.O.S., M.T. and A.M.S.; Formal analysis, H.O.S., M.T. and S.K.; Funding acquisition, F.A.-Q.; Investigation, H.O.S. and M.T.; Methodology, H.O.S., M.T., S.K. and A.M.S.; Resources, M.N. and S.K.; Software, H.O.S. and M.T.; Supervision, F.A.-Q.; Validation, F.A.-Q., S.K. and M.N.; Writing—original draft, H.O.S.; Writing—review & editing, H.O.S. and M.N. All authors have read and agreed to the published version of the manuscript.

**Funding:** The authors extend their appreciation to the Researcher Supporting Project No. (RSP-2021/73) at King Saud University, Riyadh, Saudi Arabia.

**Institutional Review Board Statement:** Not applicable.

**Informed Consent Statement:** Not applicable.

**Data Availability Statement:** Not applicable.

**Conflicts of Interest:** The authors declare that they have no conflict of interest.

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
