# Peer review of "Phytochemical Analysis of Maerua oblongifolia, and Assessment of the Genetic Stability of M. oblongifolia under In Vitro Nanoparticles Exposure"

_horticulturae, doi:10.3390/horticulturae8070610_

Round 1

Reviewer 1 Report

These may help make reading better:

1. Clean up line 37

2.line 38   ....the plant is extensively studied

3.  del space line 40

4. line 42 delete "in"

5. line 44  ...its "potential" medicinal properties

6. line 49 del space

7. line 83 delete "n" Keep A

8. line 86 ....held for 3 min, 

9. Need section describing TEM similar to GC-MS analysis section

10. Remember in section 2.3, one should describe the methods (conditions for TEM) and report on the sizes in the results section.

11. The reaction scheme should be added to this methods section

12. If the TEM micrograph is a result of this investigation, then it should be in the Results section. 

13. Line 102  .... were added   ..... media to afford 50 mL of MS in each Magenta box.  

14. Line 112- 114 should be in the results section.

15. Line 123 explain more what inter simple sequence repeat markers are?

16. Check all figure to make sure periods are in place.

17. Figures 1 and 2 should be in the Results section

18. Line 138, do spell check for past tense of program

19. Line 140   ...... was performed as follows: 

20.  Line 141  is miliq spelled correctly?

21. Remove - from line 170

22. GC-MS is poor quality; please make adjustments; sample seems to have a number of unidentified components.

23. There is major concerns about the baseline, it should be corrected and runs repeated before publishing

24. Remove bold from line 180

25. Add section in results for TEM and for Ag and Zn particles.  Zn particles seem larger than reported in the text.

26. Line 200 ....using.....

27. Ensure period is at end of sentence in lines 220 and 222

28. Line 226 be consistent with primer naming UBC?17 and UBC-21

29. Line 227 delete space

30. line 230 delete comma after where

31. line 234 remove bold and .....UBC-17 are ....

32. remove space in line 238, 239

33. show UBC-21 and UBC-20 primer to show how these differ

34. delete comma after 4 in line 245

35. delete space lines 249, 252

36. line 253 As seen; then remove bold and () 

37. Line 254 explain UPGMA 

38. Quality of figures 8 and 9 are not good

39. line 262 ....values.....

40. line 263 ...plant compared to MP while ........

41. line 268 remove bold

42. line 282 remove and

43. line 284 delete space

44. line 287 add and after ()

45. line 290 change ; to .  Then capitalize A in application

46. line 293 change enhances to enhancements

47. line 295,  add comma after ZnO NPs

48. line 296 remove comma after where

49. line 297 remove comma add . then capitalize A

50. line 300 replace able to with can

51. line 306 change word process to processes

52. line 318 place a in front of high 

53. line 322 replace treated with treatment

54. line 329 change to The obtained low polymorphisms in this study are ...

55. line 332 remove Where and make Genotoxic then replace my with may

56. line 343 - 346 sentence is too long

57. line 353 replace the nanoparticles treatment with nanoparticle treatments

58. line 330 what is Mentha x piperita?

59. Reference are in a different font style than the text

60 ref 1 is not properly cited

61. check spelling line 375

62 ref 11 is not properly cited

63 check ref 13

64. check ref 19

65. remove all capitalization of letter in ref 20

66. check ref 23, 29, 32 (herbal Medicine should be Herbal Medicine), 33 ( same problem as ref 32, 

67. Ref 38 not properly cited

68. ref 41 translate please

69. ref 47 same as comment 66

70. ref 50 change name from all cap

Author Response

lease see the attachment

Reviewer 2 Report

The authors identified the phytochemical compounds existing in M. oblongifolia leaves extract using gas chromatography and mass spectroscopy (GC-MS). They confirmed the plant growth and genetic uniformity of the plant when the plant is exposed to silver and zinc oxide nanoparticles. They found that the nanoparticles at higher concentrations induced minor genetic variation. They also confirmed that exposure of silver and zinc oxide nanoparticles to the in vitro culture media of plant tissues promote plant productions. This work advances the field of the effect of nanoparticles on plant growth. I suggest that the manuscript be accepted after the following minor revisions are made.

1.            Abstract: I think the abstract can be more concise. Now it is too detailed.

2.            Section 2.3: More details are needed to describe biosynthesis of silver nanoparticles.

3.            Section 2.4: The authors need to mention more details about the protocol of synthesizing ZnO nanoparticles.

4.            In section 3.4, the authors described Detection of Genetic uniformity Using ISSR (PCR) Molecular Marker. They need to elaborate the similarities or differences of the genetic uniformity revealed in this study and in other studies if any.

5.            Figure 8: the lines in this figure are blur. Improve the quality of this figure.

6.            In discussion section: the authors need to mention that biologically modified nanoparticles can be used for gene or drug delivery by citing relevant references (e.g., Angewandte Chemie International Edition, 2013, 125 (43), 11488-11491; Curr Top Med Chem. 2022, doi: 10.2174/1568026622666220601165005 ). Then they point out that their nanoparticles can be potentially used for the similar applications. This will improve the impact of this exciting work.

Author Response

lease see the attachment
